# Treatment of Oxidative Stress with Exosomes in Myocardial Ischemia

**DOI:** 10.3390/ijms22041729

**Published:** 2021-02-09

**Authors:** Yun Liu, Mengxue Wang, Yin Liang, Chen Wang, Keiji Naruse, Ken Takahashi

**Affiliations:** Department of Cardiovascular Physiology, Graduate School of Medicine, Dentistry, and Pharmaceutical Sciences, Okayama University, Okayama 700-8558, Japan; pdx9527@163.com (Y.L.); maoqiu985@163.com (M.W.); m13130484071@163.com (Y.L.); wangchen11228@gmail.com (C.W.); knaruse@md.okayama-u.ac.jp (K.N.)

**Keywords:** exosome, oxidative stress, exosome therapy, myocardial infarction, coronary heart disease, reactive oxygen radicals

## Abstract

A thrombus in a coronary artery causes ischemia, which eventually leads to myocardial infarction (MI) if not removed. However, removal generates reactive oxygen species (ROS), which causes ischemia–reperfusion (I/R) injury that damages the tissue and exacerbates the resulting MI. The mechanism of I/R injury is currently extensively understood. However, supplementation of exogenous antioxidants is ineffective against oxidative stress (OS). Enhancing the ability of endogenous antioxidants may be a more effective way to treat OS, and exosomes may play a role as targeted carriers. Exosomes are nanosized vesicles wrapped in biofilms which contain various complex RNAs and proteins. They are important intermediate carriers of intercellular communication and material exchange. In recent years, diagnosis and treatment with exosomes in cardiovascular diseases have gained considerable attention. Herein, we review the new findings of exosomes in the regulation of OS in coronary heart disease, discuss the possibility of exosomes as carriers for the targeted regulation of endogenous ROS generation, and compare the advantages of exosome therapy with those of stem-cell therapy. Finally, we explore several miRNAs found in exosomes against OS.

## 1. Introduction

Cardiovascular disease (CVD) has been the leading cause of mortality in recent years, and its incidence and mortality are closely related to coronary heart disease (CHD). CHD could cause a huge economic burden to regional or national medical systems [1,2]. CHD is, in fact, an inflammatory disease. Oxidative stress (OS) plays an important role in the development of coronary artery disease, and it is mainly caused by an imbalance between reactive oxygen species (ROS) production and endogenous antioxidant defense system. At low levels, ROS causes subtle changes in intracellular pathways, such as redox signal transduction, but at higher levels it causes cell dysfunction and damage [3,4,5]. In the current research on exogenous anti-OS, the effects of OS damage were mot significantly reduced [6,7]. At present, strategies for the clinical treatment and prevention of atherosclerotic CVD still focus on the pharmacotherapy of arachidonic acid metabolism and antiplatelet aggregation (platelet P2Y12 inhibitors), as well as the treatment of related risk factors, such as high blood pressure, excessive lipids, and high blood sugar [8,9,10,11,12,13].

Exosomes are small vesicles [14,15,16] that contain complex RNAs and proteins which are found in natural body fluids, including blood, saliva, urine, cerebrospinal fluid, and milk [17,18]. Discovered in 1946, exosomes were first considered as “clotting factors” [19] that improved coagulation. After 20 years, electron microscopy revealed that platelet products contain vesicles measuring 20–50 nm [20]. Until 1987, Johnstone named these vesicles as “exosomes” [21]. Exosomes can be used as carriers for intercellular communication and can regulate protein expression in receptor cells by RNA transfer [22]. Intercellular communication is necessary in maintaining tissue/organ integrity/homeostasis and inducing adaptive changes to exogenous stimuli. In response to environmental damage and pathological conditions, many cell types release various exosomes of different quality and quantity into the circulation [23,24]. During OS, the exosomes released by cells can mediate signal transduction, change the defense mechanism of receptor cells, and enhance their resistance to OS [23]. In recent years, considerable attention was paid to the important role of exosomes in CVDs, such as ischemic heart disease [25,26,27,28,29].

Exosomes are released from damaged or diseased hearts, playing an important role in disease progression [30,31,32,33]. Considering the related experiments and clinical cell therapy studies, the important roles of exosomes in myocardial injury, repair, and regeneration are being increasingly recognized. According to some studies, the Framingham risk score used to predict CVD risk correlates with circulating exosomes [34,35]. Therefore, exosomes in the circulatory system are potential biomarkers of CVD. The selective packaging of miRNAs in exosomes and their functional transfer through specific signaling molecules are also important for disease treatment [36,37]. In addition, exosomes help detect the endogenous processes of myocardial recovery, regeneration, and protection [38]. They reflect the real-time microenvironment of the lesion, indicating that they are excellent biomarkers in clinical diagnosis. Exosomes are extremely useful because they can determine the pathophysiology of heart disease noninvasively.

In ischemic myocardium, especially after reperfusion, numerous ROS are produced [39,40]. ROS directly damage tissues, inducing cell death. In transgenic mice, infarct size was found to be significantly reduced when the antioxidant protein superoxide dismutase (SOD) was overexpressed [41,42]. Increasing the level of endogenous antioxidants can prevent reperfusion injury [43,44]. Exosomes can provide precise treatment through miRNAs by selecting the corresponding target cells and manipulating the corresponding components; thus, exosomes are a powerful tool for individualized therapy and gene therapy [45,46,47]. Therefore, the upregulation of endogenous antioxidants through exosomes seems to have good prospects. Here, we discuss the potential of exosomes as carriers for regulating endogenous ROS levels to improve the course and prognosis of MI. This work also attempts to discuss the potential of exosomes as biomarkers for CVD. The advantages and disadvantages of exosomes as cell-free therapy are also examined.

## 2. Exosome-Regulated OS Responses after Myocardial Ischemia 

Although initially identified as cell debris, exosomes have many functions regulated by multiple signaling pathways. Exosomes are widely involved in the regulation of OS [48,49] and pathophysiological regulation of various cells; cellular pathophysiological processes include signal transduction, antigen presentation, and immune response [50,51]. Many of the previously conducted studies attempted to provide a detailed summary of the biogenesis of exosomes [52,53,54]. Figure 1 provides an illustration of exosome secretion under OS.

In recent years, many studies concerning CVD highlighted that exosomes not only transport proteins, RNA, DNA, and other molecules under physiological conditions but also participate in pathological conditions such as ischemia–reperfusion (I/R) injury, atherosclerosis, and cardiac remodeling [55,56,57,58,59,60,61]. They can modify gene expression and protein synthesis by inhibiting protein synthesis or initiating mRNA degradation to perform their functions at the post-transcriptional level. Moreover, circulating exosomes are considered as new biomarkers of disease performance and progression [57,62,63].

Under pathological conditions, the exosomes released during OS carry antioxidant molecules, such as superoxide dismutase 1 (SOD1) and glutathione S-transferase (GST) [62,63], and defense molecules, such as glutathione peroxidase 1 (GPX1) [64], which can be absorbed by neighboring cells to enrich their cellular defense mechanisms; thus, these cells are already protected from OS induced by adverse environmental conditions. Therefore, exosomes can potentially transfer defense molecules from one cell to another [65]. For example, in vitro, serum exosomes from healthy human volunteers attenuated H_2_O_2_-induced H9c2 cell apoptosis via ERK1/2 signaling pathway activation [66]. Moreover, cardiomyocytes (CMs) secrete miR-30a–rich exosomes after hypoxia stimulation [67]. When the release of miR-30a from exosomes is inhibited, autophagy and OS response in CMs may be maintained after hypoxia [68]. 

Furthermore, rapid ROS increase and OS occurrence are related to antioxidant depletion. Supplementation of related exogenous antioxidants, such as vitamin E and folic acid, might achieve good effects against OS [69,70]. However, a recent meta-analysis of randomized controlled trials involving 294,478 participants indicated that supplementation of exogenous vitamins and antioxidants was not associated with a reduction in the risk of major CVDs [71]. Interestingly, supplementation with *N*-acetylcysteine to increase endogenous antioxidants (e.g., glutathione (GSH)) can achieve good antioxidant capacity. After being absorbed by cells, *N*-acetylcysteine is transformed into cysteine. When the cysteine level increases, the synthesis rate of GSH also increases. More importantly, *N*-acetylcysteine supplementation not only improves the prognosis of patients but also produces no adverse side effects [72,73,74,75]. Increasing the ratio of GSH/oxidized GSH (GSSG) in patients with heart failure and acute myocardial infarction (MI) can reduce OS and improve the MI area and cardiac function [72,73,74,75]. This finding is also beneficial for the treatment of exosomes, considering that endogenous antioxidants, such as catalase, can be delivered directly through exosomes [76]. Catalase is the main enzyme that regulates H_2_O_2_ metabolism. The level of catalase gradually decreases over time after MI [77]. Its overexpression can reduce myocardial I/R injury [78]. After reformation and reshaping of exosomes by sonication and extrusion procedures, catalase could be loaded into exosomes, with a loading capacity of 20–26% [79]. In addition, catalase exosomes can be obtained by modifying parent cells (monocytes/macrophages) and then isolating from the conditioned medium [80].

As natural drug delivery nanoparticles, exosomes have the advantages of cell-based drug delivery and nanoscale size, which aid in achieving effective drug delivery. Exosomes are also lipid vesicles and ideal carriers [81]. In recent years, the application of exosomes as a biomaterial for drug delivery has improved rapidly. At present, treatment with autologous exosomes can help obtain long-term and stable activation of immune effectors [82]. Furthermore, certain drugs can modify exosomes to form carriers with different properties. In mice, the introduction of polyethylene glycol to the exosome surface significantly increased the circulation time of exosomes [83]. However, systemic delivery of exosomes appears to accumulate in the spleen and liver [84,85,86]. To solve this issue, we need to modify the exosomes to increase their targeting to specific tissues or cells. Cells that produce exosomes should be engineered to drive the expression of targeting moieties fused with exosomal membrane proteins. For example, Alvarez-Erviti et al. [87] modified dendritic cells to express Lamp2b, an exosomal membrane protein fused to neuron-specific rabies viral glycoprotein (RVG) peptide, to obtain exosomal targeting. In addition, Wiklander et al. [88] found that compared with unmodified exosomes, RVG-targeted exosomes greatly accumulate in the brain after systemic administration. At the same time, the ability of exosomes to target hypoxic cells in vivo can be enhanced by combining exosomes with hypoxia-targeting peptides or antibodies through bioengineering technology [89,90]. Moreover, exosomes released by different cells, such as immune cells, may be more effective in targeting hypoxic tissue in vivo [91]. An alternative strategy for the noninvasive targeting of magnetic drugs (i.e., enhancing drug delivery to selected tissues by applying a magnetic field gradient) was also proposed decades ago [92,93]. In this strategy, the therapeutic agent and iron oxide nanoparticles together with macrophages are incubated, leading to the production of exosomes loaded with both the therapeutic and magnetic nanoparticles. However, this method may have the disadvantages of toxicity and difficulty in targeting deep tissues. Moreover, exosomes can be used for different ways, such as intraperitoneal injection, subcutaneous injection, and nasal administration. Different administration routes may help improve the therapeutic effect [94]. For example, the intranasal administration of catalase-loaded exosomes in a mouse model of Parkinson’s disease resulted in the increased accumulation of exosomes in brain tissue after four hours [76].

Exosomes as carriers can prevent internal molecular degradation and target special tissues, thereby improving bioavailability and reducing side effects. Moreover, exosomes can serve as carriers for drug delivery and have the potential to easily manipulate the expression of RNA and proteins [87]. Exosomes naturally occur and possess adhesion proteins, which can bind to target cells and remain in target tissues during transplantation [95]. In addition, exosomes have long-term preservation and no degradation because of the existence of resistant membranes. The membrane of exosomes may pass through the blood–brain barrier [96].

Exosomes have great advantages as carriers; however, “nonvesicles,” which are distinct particles that have low electron density without restrictive membranes, are present in exosome preparation [97]. Nonetheless, the appearance of artificial nanovesicles (exosome-mimetic nanovesicles) [91,98] may be helpful in solving this issue.

## 3. Several Possible Exosomal miRNA Loads

Exosomes contain various molecules, including proteins, lipids, DNA, mRNA, and miRNA, and relevant data can be acquired from the ExoCarta database [99]. Considering the various regulatory roles of miRNA in gene expression, more attention was paid to miRNA. The proportion of miRNA in exosomes is higher than that in their parent cells [100], and miRNA can be transferred between cells through exosomes [22,36]. Meanwhile, miRNAs in exosomes are protected by vesicles and can be stably maintained in circulation; eventually, they are transferred to target cells to inhibit the expression of some genes [101,102]. For example, the knockdown of beta-secretase 1 (BACE1) mRNA and protein was detected in mouse brains after tail vein injection of siRNA-containing exosomes [87]. Therefore, miRNA seems to have a good potential as a content in exosomes.

### 3.1. MiR-19a

The abnormal expression of miRNA is related to CHD progression. MiR-19a is overexpressed in many cancer types [103,104,105]. However, information about miR-19a in CHD is limited. The plasma miR-19a level in patients with acute MI is significantly higher than that in the control group (up to 120 times), indicating a close relationship between the circulating miR-19a level and sensitivity to acute MI, with this level demonstrating high predictive and distinguishing abilities [106]. After myocardial I/R, apoptosis induced by OS is the key factor of I/R injury. Nevertheless, this damage was significantly reduced by injecting miR-19a into the myocardium of mice after MI [107]. Moreover, miR-19a derived from mesenchymal stem cells (MSC) exosomes could be delivered to the ischemic myocardium to achieve a protective effect [108]. This mechanism is mainly caused by the following ways: 1) miR-19a downregulates the expression of the target proteins in CM, phosphatase and tensin homolog (PTEN), and Bcl-2-like protein 11 (BIM), and activates the Akt and ERK signaling pathways [109]; 2) miR-19a inhibits PTEN in the heart [107,110]. However, this mechanism may not be the only way to protect myocardium. Recently, miR-19a was found to also inhibit OS-induced apoptosis by targeting the three prime untranslated region (3′UTR) of cylindromatosis (CYLD) [111]. Controlled ROS production and nuclear factor kappa-B (NF-κB) inactivation inhibit OS and regulate the expression of miR-19a, thereby inhibiting cell apoptosis induced by OS through the prevention of CYLD proliferation.

### 3.2. MiR-210

MiR-210 is a miRNA whose expression is induced and regulated by hypoxia, and it also regulates the expression of related genes [112]. MiR-210 has multiple functions because it is upregulated by hypoxia in all tested cell types and tissues [113]. In fact, miR-210 is a target of hypoxia inducible factor-1 (HIF-1) [114]. Under hypoxia, miR-210 regulates multiple cellular processes, such as inhibition of mitochondrial metabolism, promotion of mitochondrial respiration to glycolysis translocation [115], inhibition of apoptosis [116,117], and support in stem-cell survival [118,119].

HIF-1 is a transcription factor that plays an important role in cell response to a hypoxic environment [120]. The HIF-1 activity in the heart is regulated by mRNA changes and HIF-1α protein levels [121,122,123]. On the one hand, HIF-1 can mediate paracrine protection signal in the ischemic heart [124]. This protective effect can not only protect cells in ischemic regions but also potentially improve the survival of transplanted cells [125]. On the other hand, HIF-1 can reduce oxygen consumption by controlling the transcription and post-transcriptional mechanism of cells, that is, cells switch to glycolysis [126]. Under hypoxia, miR-210, which is a known miRNA regulated by HIF-1, is preferentially upregulated by HIF-1 in exosomes. One of the targets of miR-210 is iron–sulfur cluster assembly enzyme (ISCU) [115]. When the *ISCU* gene is inhibited, the mitochondrial metabolism decreases [125]. The protective effect of HIF-1 and reduced oxygen consumption can reduce ROS production and control the OS level to a certain extent in CMs. Sang-Ging et al. [125] confirmed that transplanted cardiac progenitor cells (CPCs) can resist hypoxia-induced stress in this way. In addition, endothelial cells exposed to hypoxia can produce exosome-rich miR-210 and increase CPC tolerance to OS by stimulating the PI3K/Akt pathway and other survival pathways [127].

### 3.3. MiR-133a

MiR-133a is one of the most abundant miRNAs in the heart and is crucial in the growth and development of this organ [128,129]. In fact, miR-133a is a biomarker of MI [130]. The serum level of miR-133a is significantly increased in patients with acute MI or unstable angina pectoris [131,132,133] and is closely related to the all-cause mortality rate of patients with acute coronary syndrome [132]. In MI rat models, miR-133 overexpression improved cardiac function through left ventricular ejection fraction and fractional shortening [32].

Furthermore, miR-133a is involved in the early pathology of MI and subsequent cardiac repair [134,135,136]. After hypoxia, the expression of miR-133a was found to change significantly. Overexpression of miR-133a could inhibit hypoxia-induced apoptosis and improve the ability of CMs to resist OS [137,138,139]. In I/R models, miR-133a overexpression significantly reduced CM apoptosis during OS, and this effect is likely to be mediated by targeting death-associated protein kinase 2 (DAPK2) to inhibit I/R injury [140].

MiR-133a also inhibits apoptosis in myocardial ischemic postconditioning, prevents the expression of transgelin 2 (TAGLN2) and caspase-9, and upregulates the expression of antiapoptotic protein Bcl-2 [136,139]. Interestingly, CPC can also benefit from miR-133a by reducing caspase 3 activity and targeting the proapoptotic genes *Bim* and *Bmf* [141]. At the same time, miR-133a can improve the anti-OS and survival ability of MSCs by downregulating caspase-9 and Apaf-1 expression [142]. These results indicate the therapeutic value of miR-133a in I/R injury.

## 4. Advantages of Exosome Therapy in CHD Compared with Those of Stem-Cell Therapy

Over the years, various strategies were tried to find a more effective treatment after CVD occurrence. Currently, stem-cell therapy is an attractive method for CHD prevention and treatment [143,144,145]. In 1993, Koh et al. [146] proved that skeletal muscle myoblasts can be stably transplanted into CMs, demonstrating long-term survival, proliferation, and differentiation. Recently, research focus shifted to bone marrow-derived MSCs [147], and the relevant experiments achieved favorable results [148,149,150]. Previously, differentiation characteristics were the main mechanism for cell transplantation to exert therapeutic effects, however, stem cells did not necessarily differentiate into CMs or endothelial cells after transplantation into ischemic myocardium, but the antiapoptotic, antioxidant stress, and anti-inflammatory effects were mediated by exosomes, thereby improving cardiac function after ischemia [151]. Furthermore, stem-cell transplantation can lead to arrhythmia [152,153,154,155,156,157,158].

The secretory properties of cell transplantation represent an important scientific issue in CHD. Interestingly, exosomes produced during myocardial ischemia can mediate the preventive and therapeutic effects of cell transplantation [159]. Exosomes do have a protective effect on the cardiovascular system [26,160], which was first reported more than 10 years ago [161]. In porcine and mouse models of myocardial I/R injury, 100–200 nm macromolecular complexes secreted by stem cells protected cells under OS. In the subsequent biophysical studies, the biologically active component was characterized as an exosome. More direct evidence suggested that adult stem cells repair heart tissue by releasing paracrine and autocrine factors [162]. For instance, in the isolated Langendorff I/R injury model, purified exosomes derived from MSCs reduced MI in mice [62]. In further experiments, exosome therapy restored the energy consumption and OS levels of the mouse heart within 30 min after I/R and activated cardioprotective PI3K/Akt signaling [163]. Results of a meta-analysis confirmed these cardioprotective effects of MSC-derived exosomes in myocardial injury [164]. I/R results in consumption of intracellular ATP to a large extent, and exosomes from adipose-derived stem cells (ADSCs) were observed to supplement intracellular ATP, NADH, phosphorylated AKT, and phosphorylated GSK-3β levels, while reducing phosphorylated c-JNK and recovering cell bioenergy [165]. The exosomes from ADSCs also increase IL-6 expression and phosphorylate STAT3, which in turn activates the classical signaling pathway and accelerates recovery from injury and angiogenesis after I/R [166]. The effects of exosomes derived from stem cells due to reduction of myocardial OS damage are summarized in Table 1.

Preconditioning can lead to the release of hormones or agonists, such as adenosine [167], opioids [168], and bradykinin [169]. These factors activate related signaling pathways through binding to G protein-coupled receptors, thereby activating PI3K, producing phosphoinositol, and ultimately activating downstream protein kinase B (Akt, also known as PKB or Rac) [170,171]. Davidson et al. provided an overview of exosomes, which mediate the transmission of the cardioprotective signal of remote ischemic preconditioning (RIC) and play a role in reducing OS-mediated damage [172]. Therefore, exosomes have great potential in the treatment of OS caused by MI. Similarly, a group of experiments using cardiac progenitor cell (CPC) exosomes showed that the purified CPC exosomes could be efficiently absorbed by H9c2 cells in vitro and could protect H9c2 from OS by inhibiting caspase 3/7 activation [173]. In Liu et al.’s [66] study, exosomes decreased H9c2 CM apoptosis induced by H_2_O_2_, which is the main component of OS, by activating the ERK1/2 signal pathway in vitro, thereby improving the survival rate of H9c2 cells.

Stem cells can be used to treat CVD [177]. Stem-cell therapy is currently being given the utmost research importance, but it possesses some limitations that hinder its widespread use. For example, stem-cell therapy may lead to tumor production [178,179,180]. However, the potential obstacle is their limited survival time in the heart. Implanted MSCs cannot survive for a long time (<1% of MSCs survived after one week of systemic administration) [181,182,183,184]. In the experiment of van Berlo et al., only 0.003% of transplanted cardiac stem cells differentiated into new CMs [185]. Therefore, the ability of repairing myocardial tissue in infarct sites by transplanting stem cells into CMs is limited. Additionally, considerably numerous transplanted MSCs are subsequently embedded in the vascular basement membrane of microvessels and capillaries outside the myocardial tissue [186,187,188]. However, exosomes can target specific tissues and cells through ligand-mediated targeting and magnetic drug targeting. Exosomes can home specific tissues or microenvironments according to their binding molecules. For example, integrins can bind to CMs after myocardial I/R [189] but can also home exosomes to CMs expressing ICAM1, which is a ligand for integrin.

I/R injury is an abnormal reaction of the myocardium after anoxia and reoxygenation. In myocardial I/R injury, cells cannot adapt to the rapid recovery of blood flow and oxygen level after reperfusion, causing various biochemical cascades [190]. Hence, the cells undergo repeated short nonlethal transient I/R cycles to promote biochemical adaptation to reperfusion [191,192,193,194,195]. Exosomes are relatively complex biological entities that contain various molecules, which may provide potential for treating complex injuries such as myocardial I/R injury. Various pathophysiological stress stimuli and disease conditions highly regulate the expression of proteins and RNAs in exosomes [102,196], allowing cells to produce customized exosomes with different functional characteristics. As a result, extensive biochemical and cellular activities occur in the ischemic area to correct the cascade reaction induced by ischemia and hypoxia and to prevent the occurrence of pathological conditions. Exosomes act as intercellular messengers for intercellular communication and aid in exchange of miRNA or proteins [197]. For example, after OS occurs, CPC-derived exosomes load upregulated miR-21 and enter target cells through membrane fusion, thereby inhibiting PDCD4 and cleaved caspase-3 in target cells and reducing cardiomyocyte apoptosis [175]. Moreover, during ischemia and hypoxia, the exosomes of MSC-conditioned medium can significantly reduce the nuclear OS of recipient cells through paracrine mechanisms, thereby reducing apoptosis through reduction of phospho-Smad2 and caspase-3 expression [161]. MiR-93-5p-rich exosomes from ADSCs can target the ATG7 protein through paracrine mechanisms and reduce autophagy and autophagy-related protein expression by targeting Toll-like receptor 4 (TLR4) [174]. Meanwhile, the local microenvironment of the infarct area can affect stem-cell transplantation. For example, the ischemic injury in the infarcted heart hinders the survival of transplanted stem cells, thereby reducing their beneficial effects. Proteins in exosomes secreted by the cells in the infarct area are not affected, but UV treatment can inactivate RNA function in exosomes [23]. Thus, exosome therapy is more effective than stem-cell therapy.

Unlike stem cells, exosomes can maintain their integrity during cryopreservation and transportation because they do not need to be functional. Tumor cell-derived exosomes can mediate drug resistance in chemotherapy and radiotherapy, and drug-resistant cells can transfer drug resistance to target cells through membrane particles in vitro to make these target cells resistant [198], providing cardiac protection for patients during radiotherapy and chemotherapy. At the same time, the exosomes are more stable [199] than stem cells, with no immune rejection [87]. Therefore, exosome-based therapy may be a better option than stem-cell therapy [200].

## 5. Prospects for the Clinical Application of Exosomes

The new era of the clinical application of exosomes has developed rapidly. In fact, exosomes were approved for use in some clinical trials, and exosome-based therapies are being increasingly applied in humans [201]. Exosomes are robust and stable, making them tremendously attractive as drug delivery tools. Over the past few decades, various methods were invented for loading exosomes with therapeutic drugs, indicating an extension of the drug delivery system. Exosomes may eventually be designed to have a high degree of selectivity and more effective disease targeting. The immunomodulatory and regenerative properties of exosomes are also encouraging for clinical therapy use. Exosomes may replace stem cells in treating OS-induced injury in the cardiovascular system and minimize related concerns and discussions on reducing the tumorigenicity of stem-cell transplantation. Exosome-based therapy was reported to be safe, feasible, and effective in inducing antigen-specific T-lymphocyte responses, but it still has some technical obstacles that must be overcome.

The control of the production and modification of exosomes by parental cells, as well as the quality and function of released exosomes, still require further investigation. Currently, exosomes are characterized by means of size determination and biochemical marker analysis. A more effective protocol to isolate and purify exosomes is also needed to facilitate the vesicles used in clinical research and treatment programs in order to further understand their physiological functions and link their characteristics with pathological results. Although several studies yielded significant findings on exosome therapy, most studies involved in vitro models, making it unclear whether these results reflect in vivo processes. Additionally, most of these studies simply analyzed only a select number of miRNAs or proteins. Studies rarely attempted to investigate all exosomal contents. Therefore, the available results require careful analysis and further studies are warranted in this area. The exosome separation and purification technology suffers from considerable number of limitations and fails to rule out the possibility of protein/lipoprotein contamination [202,203,204]. Moreover, available research suggests that MSCs can release a variety of different subtypes of exosomes [205], which subsequently entails further requirements for the separation and purification of exosomes. At the same time, challenges remain for the reliable tracking of exosomes in vivo, such as the penetration of nanoparticles and magnetic nanoparticles into deep tissues and the toxicity of related nanoparticles. The use of fluorescent lipophilic dyes is not suggested as they are likely to combine with the protein components of the culture medium [206]. Additionally, despite the effectiveness of CDC exosomes after intramyocardial injection, their intracoronary delivery after reperfusion remains ineffective [207]. Therefore, further research is required to elucidate an effective delivery method, such as intravenous injection or a combination of multiple delivery methods.

Currently, the conversion of exosomes to a clinical therapy is limited by the existing regulatory framework, and exosome-based therapy must still be classified. It may be classified into biomedical products or highly advanced therapy medicinal products. Therefore, formulating relevant specific treatment rules may not be necessary. Although relevant documents focusing on regulating exosome-based therapy are available [208], the possibility of requiring more detailed and special rules is not ruled out. For example, identifying, quantifying, and characterizing the main effectors that cause biological effects and determining the mode or mechanism of action may be necessary. However, achieving these tasks may take a long time. According to the existing relevant regulations, several processes still need to be accomplished before using this therapy in clinical trials (e.g., source, immunogenicity, and expected effect) [208,209].

## 6. Conclusions

Exosomes are involved in various functional behaviors and intercellular communication of cardiac cells, thereby playing an important role in the pathophysiological process of CHD. Given their relevant characteristics, exosomes are considered as therapeutic agents of OS or carriers of related drug therapies. However, the mechanism of OS still requires further exploration, and the application of exosomes into clinical treatment still need time and related experimental tests, which may require support from interdisciplinary research. Once these limitations are solved, exosomes may become a therapeutic tool for CHD.

## Figures and Tables

**Figure 1 ijms-22-01729-f001:**
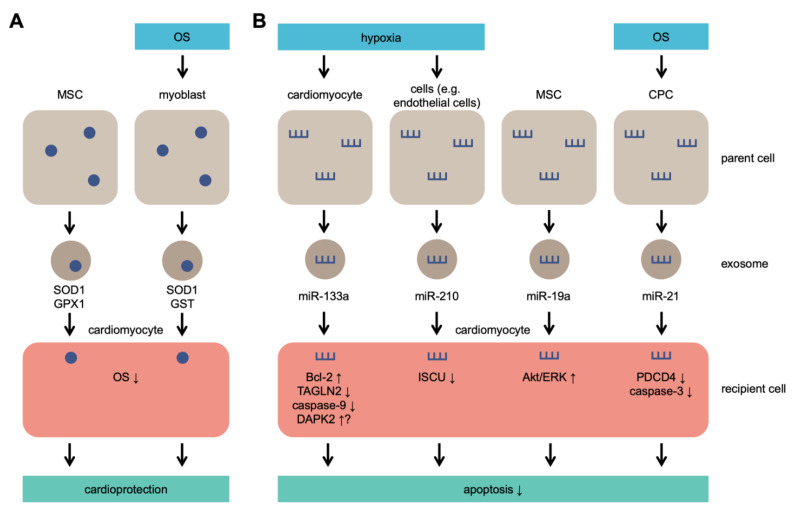
Exosome secretion under OS. Parent cells secrete exosomes containing antioxidant molecules that lead to cardioprotection (**A**) and/or microRNAs (miRs) that lead to inhibition of apoptosis (**B**). MSCs secrete SOD1, GPX1, and miR-19a without OS or hypoxia and protect cardiomyocytes exposed to OS. MSC: mesenchymal stem cells; CPC: cardiac progenitor cell; SOD1: superoxide dismutase 1; GPX1: glutathione peroxidase 1; GST: glutathione S-transferase; TAGLN2: transgelin 2; DAPK2: death-associated protein kinase 2; ISCU: iron–sulfur cluster assembly enzyme; PDCD4: programmed cell death 4.

**Table 1 ijms-22-01729-t001:** Effect of exosomes derived from stem cells on the reduction of myocardial oxidative stress (OS) damage. ADSC: adipose-derived stem cell; CPC: cardiac progenitor cell; MSC: mesenchymal stem cell; ATG7: autophagy-related 7; TLR4: Toll-like receptor 4; PDCD4: programmed cell death 4.

Origin of Exosome	Mechanistic Detail of OS Damage Reduction	References
ADSC	promotes neovascularization and alleviates inflammation and apoptosis	[165]
	upregulated miR-93-5p suppresses autophagy and inflammatory cytokine expression by targeting ATG7 and TLR4	[174]
CPC	upregulated miR-21 inhibits apoptosis by targeting PDCD4	[175]
	inhibits caspase 3/7 activity	[173]
	activates ERK1/2 pathway and inhibits apoptosis	[66]
MSC	increases ATP level and activates PI3K/Akt pathway	[163]
	activates Akt/Sfrp2 pathway	[176]
	upregulated miR-19a activates Akt/ERK pathway	[109]

## Data Availability

Not applicable.

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
