# Peer review of "Treatment of Oxidative Stress with Exosomes in Myocardial Ischemia"

_ijms, 2021, doi:10.3390/ijms22041729_

Round 1
Reviewer 1 Report
The present review describes the current knowledge regarding the role of exosomes in the regulation of oxidative stress in myocardial ischemia. Furthermore, authors discuss the advantages of exosome therapy compared with those of stem-cell therapy. However, there are major issues that must be concerned; the manuscript should be re-structured, expanded and enriched with figures and tables.
This review paper highlights the cardioprotective effects of exosomes and suggest that they could be a useful therapeutic tool to ameliorate cardiac damage during ischemia-reperfusion injury. The role of exosomes in ischemia/reperfusion injury has been reviewed and the authors should cite the following paper: Cardiovasc Res, 115(7):1156-1166, doi: 10.1093/cvr/cvy314, and point out the novelty of their review (e.g. focusing on the oxidative stress).
In the Introduction, authors should reconsider the sentence:
“At present, the strategies for the clinical treatment and prevention of atherosclerotic CVD still focus on the pharmacotherapy of arachidonic acid metabolism” since it is not accurate.
Afterwards, authors refer to some general aspects of exosomes. Please provide a figure for exosome biogenesis and exosome secretion under oxidative stress. This information is missing and it is essential to be included in order to help readers understand the therapeutic value of exosomes as a repair mechanism after myocardial injury.
In Introduction, lines 62-64, the sentence “Furthermore, the genetic information of cells in the target region can be obtained by collecting blood or urine samples without invasive biopsy. This point is a significant advantage of exosomes, especially in the human heart” should be rephrased since it does not make any sense.
At the end of this section, refer to the aim of the review.
The title of section 2 should be rephrased to Exosome-regulated oxidative stress responses after myocardial ischemia.
The section 3 entitled “Advantages of exosome therapy in CHD compared with those of stem-cell therapy” should be transferred before section 5. In this section, authors refer to the cardioprotective effects of exosomes derived from mesenchymal stem cells (MSC). Exosomes have the ability to attenuate oxidative stress and activate cardioprotective PI3K/Akt signaling after myocardial ischemia/reperfusion injury. Authors should add that meta-analysis results confirmed these cardioprotective effects of MSC-derived exosomes in myocardial injury (Zhang, H., et al. Stem Cells International, 2016, 4328362)
The authors should also present the role of exosomes with different origin on modulating the oxidative stress after myocardial ischemia. They should incorporate some relevant publications that are missing such us
Xiao X, et al. Cell Death & Disease. 7, e2277 (2016) and Liu Zhi, et al. Cell Death Discovery. 5, 79 (2019). Also, they should provide a table presenting the origin of exosomes and how they reduce the oxidative stress after myocardial ischemia (mechanistic details).
In part 5 entitled “Prospects for the clinical application of exosomes”, authors state that exosomes have been approved for use in some clinical trials, and the exosome-based therapies have been increasingly applied in humans. Authors provide a publication from 2006 describing the preclinical data, Phase I trials and optimization protocols for improving their immunogenicity in tumor bearing patients. Authors should reconsider this sentence and provide a paragraph with all the limitations of using exosomes in clinical practice. In addition to appropriate isolation methods, dosage and routes of administration for exosomes remain controversial. Furthermore, most studies simply analyzed only a select number of miRNAs or proteins, without fully covering all the differentially expressed exosomal contents. I believe that this paragraph is essential to help reader understand why to date, the possible roles of exosomes in cardiac diseases have not yet been completely elucidated in terms of routine clinical practice.
In the manuscript, the abbreviations must be explained in their first appearance in the text and please avoid repetitions.
Author Response
We thank the reviewer for their generous comments on the manuscript and have edited the manuscript to address her/his concerns.
The role of exosomes in ischemia/reperfusion injury has been reviewed and the authors should cite the following paper: Cardiovasc Res, 115(7):1156-1166, doi:10.1093/cvr/cvy314, and point out the novelty of their review (e.g. focusing on the oxidative stress).
Answer:
Thank you for suggesting an important earlier literature. We cited the paper and mentioned it in the text.
Line 295.
Davidson et al. provided an overview of exosomes, which mediate the transmission of the cardioprotective signal of remote ischemic preconditioning (RIC), and it plays a role in reducing OS-mediated damage [174].
”At present, the strategies for the clinical treatment and prevention of atherosclerotic CVD still focus on the pharmacotherapy of arachidonic acid metabolism”. This sentence should be corrected, in order to be more accurate.
Answer:
In accordance with the reviewer’s suggestion, we corrected the sentence.
Line 35.
At present, the strategies for the clinical treatment and prevention of atherosclerotic CVD still focus on the pharmacotherapy of arachidonic acid metabolism and antiplatelet aggregation (platelet P2Y12 inhibitors), as well as the treatment of related risk factors such as high blood pressure, excessive lipids, and high blood sugar [8-13].
Authors should provide a figure for exosome biogenesis and exosome secretion under oxidative stress.
Answer:
Thank you for your valuable suggestion. We added Figure 1 that shows exosome secretion under oxidative stress. About the exosome biogenesis, we found excellent earlier literatures. Therefore, we cited the papers in the text.
Line 85.
Many of the previously conducted studies have attempted to provide a detailed summary of the biogenesis of exosomes [52-54]. Figure 1 provides an illustration of exosome secretion under OS.
The sentence “Furthermore, the genetic information of cells in the target region can be obtained by collecting blood or urine samples without invasive biopsy. This point is a significant advantage of exosomes, especially in the human heart” should be rephrased since it does not make any sense.
Answer:
We rephrased this sentence as follows:
Line 65.
They reflect the real-time microenvironment of the lesion, indicating as excellent biomarkers in clinical diagnosis. Exosomes are extremely useful because they can determine the pathophysiology of heart disease noninvasively.
The title of section 2 should be rephrased to Exosome-regulated oxidative stress responses after myocardial ischemia.
Answer:
The title of section 2 is changed to “Exosome-regulated oxidative stress responses after myocardial ischemia” in accordance with the reviewer’s comment.
The section 3 entitled “Advantages of exosome therapy in CHD compared with those of stem-cell therapy” should be transferred before section 5. In this section, authors refer to the cardioprotective effects of exosomes derived from mesenchymal stem cells (MSC). Exosomes have the ability to attenuate oxidative stress and activate cardioprotective PI3K/Akt signaling after myocardial ischemia/reperfusion injury. Authors should add that meta-analysis results confirmed these cardioprotective effects of MSC-derived exosomes in myocardial injury (Zhang, H., et al. Stem Cells International, 2016, 4328362).
Answer:
We transferred section 3 before section 5. Also, we added the meta-analysis literature suggested by the reviewer in the text:
Line 281.
Results of a meta-analysis have confirmed these cardio-protective effects of MSC-derived exosomes in myocardial injury [166].
The authors should also present the role of exosomes with different origin on modulating the oxidative stress after myocardial ischemia. They should incorporate some relevant publications that are missing such us Xiao X, et al. Cell Death & Disease. 7, e2277 (2016) and Liu Zhi, et al. Cell Death Discovery. 5, 79 (2019). Also, they should provide a table presenting the origin of exosomes and how they reduce the oxidative stress after myocardial ischemia (mechanistic details).
Answer:
Now we add Table 1. Effect of exosomes derived from stem cells on the reduction of myocardial OS damage.
In part 5 entitled “Prospects for the clinical application of exosomes”. Authors should reconsider this sentence and provide a paragraph with all the limitations of using exosomes in clinical practice. In addition to appropriate isolation methods, dosage and routes of administration for exosomes remain controversial. Furthermore, most studies simply analyzed only a select number of miRNAs or proteins, without fully covering all the differentially expressed exosomal contents. I believe that this paragraph is essential to help reader understand why to date, the possible roles of exosomes in cardiac diseases have not yet been completely elucidated in terms of routine clinical practice.
Answer:
Thank you for your valuable suggestion. We substantially added descriptions about limitations on the clinical use of exosomes.
Line 380.
Although several studies have yielded significant findings on exosome therapy, most studies involved in vitro models, making it unclear whether these results reflect in vivo processes. Additionally, most of these studies have simply analyzed only a select number of miRNAs or proteins. Rarely has any study attempted to investigate all exosomal contents. Therefore, the available results require careful analysis and further studies are warranted in this area. The exosome separation and purification technology suffers from considerable number of limitations and fails to rule out the possibility of protein/lipoprotein contamination [204-206]. Moreover, available research suggests that MSCs can release a variety of different subtypes of exosomes [207], which subsequently entails further requirements for separation and purification of exosomes. At the same time, there remain challenges for reliable tracking of exosomes in vivo, such as the penetration of nanoparticles and magnetic nanoparticles into deep tissues and the toxicity of related nanoparticles. The use of fluorescent lipophilic dyes is not suggested as they are likely to combine with the protein components of the culture medium [208]. Additionally, despite the effectiveness of CDC exosomes after intramyocardial injection, their intracoronary delivery after reperfusion remains ineffective [209]. Therefore, further research is required for elucidating an effective delivery method, such as intravenous injections or a combination of multiple delivery methods.
Avoid duplication of abbreviations.
Answer:
Thank you for your suggestion. We removed the duplication of abbreviations.
Reviewer 2 Report
Global comments:
This review analyses the causes producing Oxidative Stress in patients with Myocardial Infraction, and the regulation possibilities by natural anticoagulants, which can be enhanced by using exosomes, as antioxidant drugs are ineffective for this objective. I congratulate the authors for this clear and extensive review and their presentation and discussion of the new treatment possibilities that they anticipate, with the promising use of exosomes wrapped with miRNAs or proteins, thus providing the expected beneficial activities locally, where disease is active, and where pathological damages are present. This report is well-written and very informative, although it can be understood a bit speculative for new therapies. But the authors duly present the treatment possibilities and limitations, and they contribute to open new areas of investigation.
Minor comments:
I suggest to add 1 or 2 tables and 1 or 2 figures for rendering this review more attractive and more friendly to read. For example a table can report what is responsible for the Reactive Oxidative Species as compared to the natural antioxidant molecules and mechanisms. A figure can also present the various exosomes/Extracellular Vesicles, which can be released before and during the Myocardial Infraction event, or during clot removal, and the proposed exosomes for fighting the Oxidative Stress.
References are abundant and correctly indexed.
Author Response
We thank the reviewer for her/his generous comments on the manuscript and have edited the manuscript to address her/his concerns.
A figure can also present the various exosomes/Extracellular Vesicles, which can be released before and during the Myocardial Infraction event, or during clot removal, and the proposed exosomes for fighting the Oxidative Stress.
Answer:
Thank you for your valuable suggestion. In accordance with the suggestion, we added Figure 1 and Table 1.
Figure 1. Exosome secretion under OS
Table 1. Effect of exosomes derived from stem cells on the reduction of myocardial OS damage
Round 2
Reviewer 1 Report
The authors improved their manuscript. I have no further comments